# A SARS-CoV-2 Infection High-Uptake Program on Healthcare Workers and Cancer Patients of the National Cancer Institute of Naples, Italy

**DOI:** 10.3390/healthcare10020205

**Published:** 2022-01-20

**Authors:** Anna Crispo, Piergiacomo Di Gennaro, Sergio Coluccia, Sara Gandini, Concetta Montagnese, Giuseppe Porciello, Flavia Nocerino, Maria Grimaldi, Mariangela Tafuri, Assunta Luongo, Emanuela Rotondo, Alfonso Amore, Francesco Labonia, Serena Meola, Stefanie Marone, Giovanni Pierro, Simona Menegozzo, Leonardo Miscio, Francesco Perri, Maurizio Rainisio, Attilio A. M. Bianchi, Ernesta Cavalcanti, Marco Cascella, Egidio Celentano

**Affiliations:** 1Epidemiology and Biostatistics Unit, Istituto Nazionale per lo Studio e la Cura dei Tumori, “Fondazione Pascale”—IRCCS, 80131 Naples, Italy; a.crispo@istitutotumori.na.it (A.C.); piergiacomo.digennaro@istitutotumori.na.it (P.D.G.); sergio.coluccia@istitutotumori.na.it (S.C.); g.porciello@istitutotumori.na.it (G.P.); f.nocerino@istitutotumori.na.it (F.N.); m.grimaldi@istitutotumori.na.it (M.G.); mariangela.tafuri@istitutotumori.na.it (M.T.); assunta.luongo@istitutotumori.na.it (A.L.); e.rotondo@istitutotumori.na.it (E.R.); e.celentano@istitutotumori.na.it (E.C.); 2Department of Experimental Oncology, European Institute of Oncology (IEO)—IRCCS, 20141 Milan, Italy; sara.gandini@ieo.it; 3Division of Surgery of Melanoma and Skin Cancer, Istituto Nazionale per lo Studio e la Cura dei Tumori, “Fondazione Pascale”—IRCCS, 80131 Naples, Italy; a.amore@istitutotumori.na.it; 4Laboratory Medicine Unit, Istituto Nazionale per lo Studio e la Cura dei Tumori, “Fondazione Pascale”—IRCCS, 80131 Naples, Italy; francesco.labonia@istitutotumori.na.it (F.L.); serena.meola@istitutotumori.na.it (S.M.); stefanie.marone@istitutotumori.na.it (S.M.); giovanni.pierro@istitutotumori.na.it (G.P.); e.cavalcanti@istitutotumori.na.it (E.C.); 5Medical Direction, Istituto Nazionale per lo Studio e la Cura dei Tumori, “Fondazione Pascale”—IRCCS, 80131 Naples, Italy; s.menegozzo@istitutotumori.na.it (S.M.); leonardo.miscio@istitutotumori.na.it (L.M.); 6Head and Neck Medical and Experimental Oncology Unit, Istituto Nazionale per lo Studio e la Cura dei Tumori, “Fondazione Pascale”—IRCCS, 80131 Naples, Italy; f.perri@istitutotumori.na.it; 7AbaNovus srl, 18038 Sanremo, Italy; Maurizio.Rainisio@AbaNovus.com; 8Directorate-General for Management, IRCCS Fondazione G. Pascale, 80131 Naples, Italy; a.bianchi@istitutotumori.na.it; 9Division of Anesthesia and Pain Medicine, Istituto Nazionale Tumori, IRCCS Fondazione G. Pascale, 80131 Naples, Italy; m.cascella@istitutotumori.na.it

**Keywords:** SARS-CoV-2, COVID-19, healthcare workers, surveillance program, cancer patients

## Abstract

Background: From the beginning of 2020, Severe Acute Respiratory Syndrome Coronavirus 2 (SARS-CoV-2) quickly spread worldwide, becoming the main problem for the healthcare systems. Healthcare workers (HCWs) are at higher risk of infection and can be a dangerous vehicle for the spread of the virus. Furthermore, cancer patients (CPs) are a vulnerable population, with an increased risk of developing severe and lethal forms of Coronavirus Disease 19 (COVID-19). Therefore, at the National Cancer Institute of Naples, where only cancer patients are treated, a surveillance program aimed to prevent the hospital access of SARS-CoV-2 positive subjects (HCWs and CPs) was implemented. The study aims to describe the results of the monitoring activity for the SARS-CoV-2 spread among HCWs and CPs, from March 2020 to March 2021. Methods: This surveillance program included a periodic sampling through nasopharyngeal molecular swabs for SARS-CoV-2 (Real-Time Polymerase Chain Reaction, RT-PCR). CPs were submitted to the molecular test at least 48 h before hospital admission. Survival analysis and multiple logistic regression models were performed among HCWs and CPs to assess the main SARS-CoV-2 risk factors. Results: The percentages of HCWs tested with RT-PCR for the detection of SARS-CoV-2, according to the first and the second wave, were 79.7% and 91.7%, respectively, while the percentages for the CPs were 24.6% and 39.6%. SARS-CoV-2 was detected in 20 (1.7%) HCWs of the 1204 subjects tested during the first wave, and in 127 (9.2%) of 1385 subjects tested in the second wave (*p* < 0.001); among CPs, the prevalence of patients tested varied from 100 (4.6%) during the first wave to 168 (4.9%) during the second wave (*p* = 0.8). The multivariate logistic analysis provided a significant OR for nurses (OR = 2.24, 95% CI 1.23–4.08, *p* < 0.001) compared to research, administrative staff, and other job titles. Conclusions: Our findings show that the positivity rate between the two waves in the HCWs increased over time but not in the CPs; therefore, the importance of adopting stringent measures to contain the shock wave of SARS-CoV-2 infection in the hospital setting was essential. Among HCWs, nurses are more exposed to contagion and patients who needed continuity in oncological care for diseases other than COVID-19, such as suspected cancer.

## 1. Introduction

Italy was the first European state to be seriously affected by the Severe acute respiratory syndrome coronavirus 2 (SARS-CoV-2) pandemic [1]. Remarkably, in the Italian peninsula, from the beginning of the pandemic to December 2020, 1,757,394 cases and 61,240 deaths were recorded; of which 167,433 cases and 2064 deaths occurred in the Campania region (South Italy) [1,2]. Thus, since the World Health Organization (WHO) declared Coronavirus Disease 19 (COVID-19) as a pandemic (11 March 2020), drastic preventive measures to contain the viral contagion, such as social distancing and confinement, and use of personal protective equipment were immediately adopted. 

In the first phase of the pandemic, between February and June 2020, the epidemiological surveillance focused mainly on symptomatic or suspected cases or contacts with SARS-CoV-2 positives [3]. Later, in the second pandemic wave between September and December 2020, the epidemiological surveillance addressed different population groups including students, teachers, school workers, healthcare workers (HCWs), and patients who needed hospital treatment for diseases other than COVID-19, such as suspected cancer. This surveillance was achieved through serological tests and nasopharyngeal swabs. Among these categories, maximum attention was paid to HCWs since they can represent an important vehicle for the spread of the virus, especially among vulnerable populations, such as cancer patients who are at increased risk of becoming ill with more serious symptoms of SARS-CoV-2 infection [4,5,6,7,8,9]. In Italy, starting from March 2020, the National Cancer Institute of Naples (Istituto di Ricovero e Cura a Carattere Scientifico—IRCCS, Fondazione Pascale) promoted a surveillance program for HCWs and cancer patients admitted to our hospital [10]. 

A careful surveillance program that pays particular attention to both fragile patients and workers can result in different care realities. The aim is to structure a validated pathway that, through adaptations, can be useful in different contexts. In a particularly changing pandemic reality, the strengthening of preventive measures, associated with vaccination campaigns, remains the best strategy to safeguard health services already severely tested.

In this study, the results of this high-uptake program are presented.

## 2. Materials and Methods

### 2.1. Study Design

This is an observational prospective study aimed at describing the prevalence of SARS-CoV-2 infection detected by molecular nasal and oropharyngeal swab RT-PCR testing in HCWs and cancer patients (CPs). These populations were followed from March 2020 to March 2021. According to the course of the SARS-CoV-2 epidemic in Italy [2,3], different periods (pandemic waves and transition phases) have been considered. In particular, the study period encompassed:The first wave, from 4 March 2020, to 31 May 2020;A transition phase, from 1 June 2020, to 27 September 2020;The second wave from 28 September 2020, to 3 January 2021;The second transition phase from 4 January 2021, to 4 April 2021.

This study was approved by the Ethics Committee of the National Cancer Institute of Naples (IRCCS, Fondazione Pascale) (number 61/20).

### 2.2. Sample

This study involved all HCWs of the National Cancer Institute of Naples (IRCCS, Fondazione Pascale) who agreed to participate in the surveillance program (signed informed consent obtained) and all CPs who had to get a medical examination or had to be hospitalized in the Institute from 4 March 2020 to 4 April 2021. 

For HCWs, the surveillance program included a periodic sampling through nasopharyngeal molecular swabs for SARS-CoV-2. The tests were performed in HCWs who were asymptomatic; symptomatic HCWs and those who refused to be tested were excluded. A subject was defined as symptomatic if any of the following symptoms/signs were manifested: fever, cough, dyspnea, chills, anosmia, ageusia. 

SARS-CoV-2 positive HCWs did not have access to the hospital and were referred to the territorial healthcare system for testing and contact tracing. HCWs who were in contact with COVID-19 patients or SARS-CoV-2 infected individuals were included in a quarantine program by the territorial healthcare system. They were excluded from working activities until the favorable opinion of the health authorities. 

According to our SARS-CoV-2 infection prevention policy, the pathways of CPs who had to be admitted to hospital for scheduled hospitalization, procedures or visits, had to carry out the molecular SARS-CoV-2 diagnostic test within 48 h before hospitalization. For each positive test, the local health authorities were alerted to manage the positive CPs and their contacts. They were referred to multispecialty centers for COVID-19 patients (Figure 1).

### 2.3. Tests Used for SARS-CoV-2

The Laboratory Medicine Unit of the National Cancer Institute of Naples (IRCCS, Fondazione Pascale) has been involved since the beginning of the program of health surveillance for all HCWs, as well as patients in the identification of SARS-CoV-2 infection. As recommended by the WHO, molecular testing has been used as the reference method for the identification of SARS-CoV-2 infectious cases [11]. Nucleic acids extraction and subsequent Real-Time PCR (RT-PCR) detection of SARS-CoV-2 RNA from nasopharyngeal swabs were performed [12]. The Charité algorithm (Berlin, Germany) worked out by Corman et al. [13], which is based on RT-PCR SARS-CoV-2 detection of E and RdRp genes, was used as a reference method.

### 2.4. Statistical Analysis

The SARS-CoV-2 prevalence was calculated for each epidemic period (1st wave, 1st transition phase, 2nd wave, 2nd transition phase) and was reported according to the age, gender, department and job title of HCWs and the age, gender, type and number of accesses of patients; HCWs or CPs who had multiple tests in the same wave were taken into account and the repeated measures were excluded.

A univariate analysis was implemented for the 2nd wave only, according to HCWs’ features and CPs through the Chi-Square test. Adjusted Odds Ratios (ORs) and 95% Confidence Intervals (CIs) were estimated on the 2nd wave by unconditional multiple logistic regression model with terms of age (≤44, >44), gender, job title (ancillary services, non-medical-area, nurse, physician, research staff, administrative staff, technologist and other), department (clinical care, surgery, research, administrative and operational service), and molecular swabs (1–3, 4–5, 6–11, not present in table) to assess the main SARS-CoV-2 risk factors. In the second pandemic wave, a Cumulative Hazard Function by CPs’ accesses and HCWs’ job title was calculated; *p*-values were produced by the Log-rank test. Hazard Ratios and 95% CIs were calculated through the Cox-model. The models were adjusted by age and gender. All analyses were performed using R software (version 4.0.2, R Core Team, Vienna, Austria).

## 3. Results

A total of 1510 HCWs were tested with molecular nasal and oropharyngeal swab RT-PCR testing for the detection of SARS-CoV-2 [10]. 1204 (79.7%) and 1385 (91.7%) of HCWs were tested during the first and the second waves, respectively. A total of 8733 CPs, who were admitted to the National Cancer Institute of Naples (IRCCS, Fondazione Pascale), were tested. A total of 2152 (24.6%) and 3462 (39.6%) of CPs were tested during the first and second waves, respectively. The total number of molecular swabs executed over the study period were 12,677 for HCWs and 15,155 for CPs. SARS-CoV-2 was detected in 20 (1.7%) HCWs during the 1st wave, and 127 (9.2%) in the second wave (*p* < 0.001); while among CPs, the prevalence varied from 100 (4.6%) during the first wave to 168 (4.9%) during the second wave (*p* = 0.8) (Table 1).

The highest prevalence among HCWs was detected for the second wave: 127 positives of 1385 HCWs (9.2%) with 10.8% being over 44 years of age and 10.3% were male.

Amongst HCWs, Table 2 shows the distribution of positive to SARS-CoV-2 for Departments and Job Title. The highest incidence rate was observed for nurses (14.9%).

Among CPs, the prevalence during the first wave was similar to that observed in the second (4.6% and 4.9%, respectively); the number of positive patients increased with the number of accesses during the second wave (2.1% one access 17.7% > 4 accesses) (Table 3).

Table 4 shows the associations between SARS-CoV-2 positive HCWs and features during the second wave. The univariate analysis showed that there were significant associations between virus positivity and age and job title. Notably, a statistically significant risk to be infected (OR = 2.24 95% CI 1.23–4.08) was observed in nurses compared to research/administrative staff.

Table 5 illustrates the associations between SARS-CoV-2 positive CPs and features during the second wave.

The univariate analysis reports that there was a significant association with the number of accesses. This finding was confirmed in the multivariate analysis where the risk categories: two, three and four or more accesses were statistically significant compared to one access (ORs = 3.76, 9.38, and 10.12, respectively).

Figure 2 shows the statistically significant infection cumulative hazard function either for CPs’ access (A) *p* < 0.001 or HCWs’ job title (B) *p* < 0.001.

The Cox multivariate analysis demonstrates a significant risk for CPs’ access to be infected that increase from 4.13 for two access to 21.04 for four access and more (see Table 6).

The Cox multivariate analysis also confirms the significant risk for nurses (HR = 2.05 95% CI 1.16–3.64) (Table 7).

## 4. Discussion

Health surveillance on HCWs and patients represents a key strategy to strengthen and protect a workforce and the weakest and most compromised individuals, such as hospitalized patients [14]. Once transmission between asymptomatic subjects was known [15], periodic health surveillance among asymptomatic or mild-symptomatic HCWs helped to reduce the risk of infection to colleagues, their families, and patients in the workplace [14,16]. During the first phase of the SARS-CoV-2 pandemic, several guidelines were developed by the WHO, the European Centre for Disease Prevention and Control (ECDC), and the European Agency for Safety and Health at Work (EU-OSHA) to reduce SARS-CoV-2 risk of infection in HCWs; however, the HCWs could themselves become part of risk reduction strategies, through direct participation in all activities related to risk mitigation, monitoring and facilitating the reintegration of other workers previously affected by SARS-CoV-2. Furthermore, the HCWs involvement cannot be separated from the organization of “Health surveillance” with the purpose of planning adequate protection measures for fragile categories of workers [17]. Hospitalized cancer patients require ongoing evaluation and immediate treatment even during a pandemic. Nevertheless, since these patients are primarily immunocompromised, they are at greater risk of serious COVID-19 [18]. 

Our work shows the results of a surveillance program on HCWs and patients of an oncological facility in Campania (Italy). Notably, a higher incidence of infections among HCWs than in the general population was observed [19]. This finding confirmed the results of a previous prospective study that demonstrated an increased SARS-CoV-2 infection prevalence among HCWs (7.3%) compared to non-HCWs (0.4%) [20]. Several studies reported a considerable risk of contracting SARS-CoV-2 infection among HCWs [19,20,21,22,23], even if the incidence of severe disease and deaths was significantly low [24].

To plan a safe working environment and to maintain effective healthcare services, many studies focused on the risks associated with specific roles [25,26]. Investigations have suggested that HCWs at increased risk include porters, cleaners, healthcare-assistants, therapists, junior clinicians, but mostly nurses [27,28]. In line with the results of other studies, in our investigation, nurses were the largest group among HCWs to be infected. These HCWs are more prone to care for patients with suspected or confirmed COVID-19, mainly in the hospital with a higher proportion of patients with COVID-19 [20]. In addition to the high risk of SARS-CoV-2 infection for healthcare professionals, it has been seen that they have experienced several mental problems, such as anxiety, stress, and burnout syndrome, a further problem to the already complicated working conditions in the pandemic phase. This could suggest the design of ad hoc programs for the prevention of mental disorders alongside health surveillance [29,30,31,32,33]. In support of this, it was shown that work overload and stress represented risk factors in contracting infectious diseases; in particular, this risk increased as the number of working hours increased, exposing HCWs for a longer time to patients, viruses, and mental stress [34]. 

Our findings show that the positivity rate between the two waves in the HCWs increased over time but not in the CPs. Therefore, the importance of adopting stringent measures to contain the shock wave of SARS-CoV-2 infection in the hospital setting was essential. Among HCWs, nurses are more exposed to contagion and patients who needed continuity in oncological care for diseases other than COVID-19, such as suspected cancer. 

Notably, for CPs, the risk of infection increases with the number of accesses. It is necessary, therefore, to structure safe care pathways and organize, where possible, ad hoc methods for the provision of care. Our findings, for example, support a telemedicine-based approach for frail patients. This strategy represents a great opportunity to facilitate continued assistance for patients improving their access to care. Furthermore, telemedicine is cost-effective and can facilitate patient-centered treatments [35]. At the Cancer Institute, this model has been provided since the first COVID-19 wave [9] and will also be implemented in the post-pandemic era [36]. 

The major limitation of our study is that it was conducted in the pre-vaccinal era; however, the data that emerged suggest the importance of information and education programs [37]. Another limitation is that it is a single-center analysis: consequently, results may not be applicable to different healthcare settings. Nevertheless, since the beginning of the pandemic, our cancer center was declared ‘COVID-free’, and diagnosed or suspected COVID-19 patients were not admitted. In particular, CPs who were SARS-CoV-2 positive or symptomatic COVID-19 patients were referred to multispecialty centers for COVID-19 patients. Thus, all HCWs had the same risk of becoming infected in the hospital setting. Presumably, the nurses contracted the infection in an out-of-hospital setting. Indeed, during the most hectic phases of the pandemic, many nurses were recruited for community assistance programs.

## 5. Conclusions

Our findings underline that the adoption of stringent measures has been essential to contain the shock wave of SARS-CoV-2 infection in the hospital setting. In a far-sighted way, these measures involved both staff and patients.

We found that the positivity rate between the two waves in the HCWs increased over time, but not in the CPs. Additionally, our results suggest the adoption of safety measures in the HCWs, even if this is not directly related to the positivity of the CPs.

Among HCWs, nurses are at greater risk than administrative staff. Our findings clearly recommend the constant use of individual protection devices for nurses and that they undergo COVID-19 surveillance, particularly more frequently for those who participate in community assistance programs.

## Figures and Tables

**Figure 1 healthcare-10-00205-f001:**
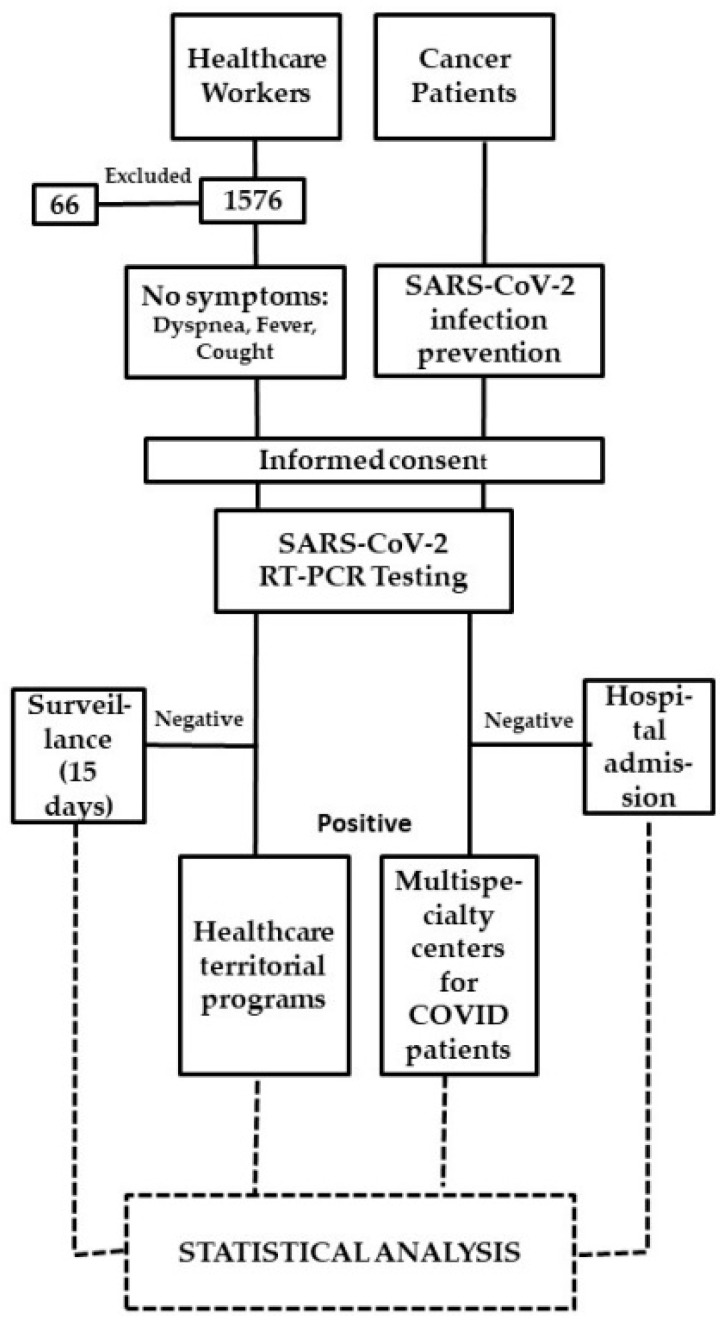
Study flow chart.

**Figure 2 healthcare-10-00205-f002:**
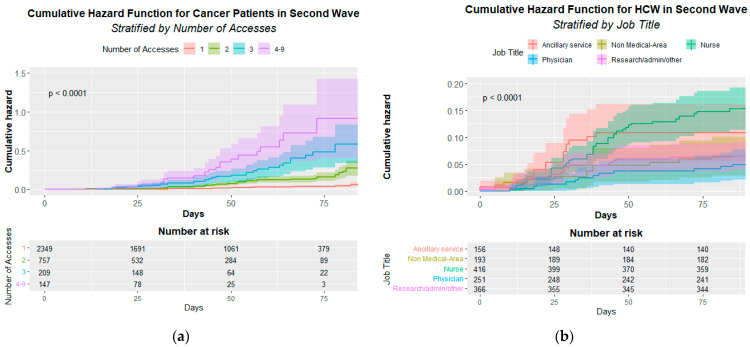
Cumulative Hazard Function for cancer patients (**a**) and Healthcare Workers (**b**). (**a**) Infection Cumulative Hazard Function by patient’s accesses in the second pandemic wave. The *p*-value of the Log-rank test is at the top left. (**b**) Infection Cumulative Hazard Function by HCWs’ job title (Healthcare Workers’ Job Title) in the second pandemic wave. The *p*-value of the Log-rank test is at the top left.

**Table 1 healthcare-10-00205-t001:** SARS-CoV-2 incidence rate of Healthcare Workers (HCWs, Overall *N* = 5106 swabs for 1510 healthcare workers, multiple tests in the same wave are excluded) and Cancer Patients (CPs; Overall *N* = 10,846 accesses for 8733 patients, multiple accesses in same wave are excluded) from the first to the second wave of the pandemic.

HCWs and CPs	First Wave: March–May 2020			1^ Transition Phase June–September 2020			Second Wave October–December 2020			2^ Transition Phase January–March 2021		
	HCWs	CPs		HCWs	CPs		HCWs	CPs		HCWs	CPs	
Total (%) ^1^	1204 (79.7%)	2152 (24.6%)		1268 (84.0%)	1793 (20.5%)		1385 (91.7%)	3462 (39.6%)		1249 (82.7%)	3439 (39.4%)	
Positive (%)	20 (1.7%)	100 (4.6%)		3 (0.2%)	14 (0.8%)		127 (9.2%)	168 (4.9%)		9 (0.7%)	40 (1.2%)	
	**Positive/TOT** **(%)**		***p*-Value ^2^**	**Positive/TOT** **(%)**		***p*-Value ^2^**	**Posivtive/TOT** **(%)**		***p*-Value ^2^**	**Positive/TOT** **(%)**		***p*-Value ^2^**
**Age**												
≤44	13/551 (2.4%)	8/269 (3.0%)	0.77	3/612 (0.5%)	5/261 (1.9%)	0.10	51/682 (7.5%)	28/530 (5.3%)	0.16	10/621 (1.6%)	10/615 (1.6%)	1
>44	7/653 (1.1%)	92/1883 (4.9%)	**<0.001**	0/656	9/1532 (0.6%)	0.11	76/703 (10.8%)	140/2932 (4.8%)	**<0.001**	2/625 (0.3%)	30/2824 (1.1%)	0.13
**Gender**												
Male	6/588 (1.0%)	40/992 (4.0%)	**0.001**	1/566 (0.2%)	6/840 (0.7%)	0.31	66/638 (10.3%)	83/1683 (4.9%)	**<0.001**	8/556 (1.4%)	24/1717 (1.4%)	1
Female	14/616 (2.3%)	60/1160 (5.2%)	**0.005**	2/702 (0.3%)	8/953 (0.8%)	0.26	61/747 (8.2%)	85/1779 (4.8%)	**0.001**	4/693 (0.6%)	16/1722 (0.9%)	0.54

^1^ Total percentage represented the total number of HCWs and CPs included in the analysis. ^2^ Pearson’s Chi-square Test at significance level of 95%. Bold numbers indicate statistically significant results.

**Table 2 healthcare-10-00205-t002:** SARS-CoV-2 incidence rate of Healthcare Workers (HCWs; *N* = 1510) from the first to the second wave of the pandemic.

Healthcare Workers (HCWs)	First Wave March–May 2020	1^ Transition Phase June–September 2020 ^2^	Second Wave October–December 2020 ^3^	2^ Transition Phase January–March 2021 ^4^
**Departments**				
Clinical care	8/403 (2.0%)	0/432	46/444 (10.4%)	2/418 (0.5%)
Surgery	11/293 (3.8%)	1/315 (0.3%)	34/332 (10.2%)	4/305 (1.3%)
Research	0/206	1/240 (0.4%)	16/248 (6.5%)	2/234 (0.9%)
Administrative	0/150	1/175 (0.6%)	14/203 (6.9%)	0/183
Operational Services	1/152 (0.7%)	0/105	17/156 (10.9%)	1/105 (1.0%)
**Job Title**				
Ancillary services ^1^	1/152 (0.7%)	0/105	17/156 (10.9%)	1/105 (1.0%)
Non medical-area	1/173 (0.6%)	0/185	13/193 (6.7%)	2/189 (1.1%)
Nurse	4/356 (1.1%)	1/387 (0.3%)	62/416 (14.9%)	4/377 (1.1%)
Physician	7/238 (2.9%)	1/247 (0.4%)	12/251 (4.8%)	0/233
Research staff	7/205 (3.4%)	0/242	19/252 (7.5%)	2/236 (0.8%)
Techno/Administr. Staff/Other	0/80	1/101 (1.0%)	4/114 (3.5%)	0/105
**Molecular Swabs**				
Total (%)	1877 (14.8%)	2374 (18.7%)	5951 (46.9%)	2475 (19.5%)
Mean (SD)	1.56 (1.09)	1.87 (0.85)	4.30 (1.70)	1.98 (1.28)

^1^ Ancillary services consisting of cleaners, security guards. SD, standard deviation. ^2^ One missing value for Departments and Job Title. ^3^ Two missing values for Departments and three for Job Title. ^4^ Four missing values for Departments and Job Title.

**Table 3 healthcare-10-00205-t003:** SARS-CoV-2 incidence rate for Cancer Patients (CPs; *N* = 8733) from the first to second wave of the pandemic.

Cancer Patients	First Wave: March–May 2020	1^ Transition Phase June–September 2020	Second Wave October–December 2020	2^ Transition Phase January–March 2021
Number of accesses				
One	56/1562 (3.6%)	7/1338 (0.5%)	50/2349 (2.1%)	22/2741 (0.8%)
Two	33/494 (6.7%)	3/344 (0.9%)	57/757 (7.5%)	10/500 (2.0%)
Three	8/78 (10.3%)	2/76 (2.6%)	35/209 (16.7%)	3/102 (2.9%)
Four or more	3/18 (16.7%)	2/35 (5.7%)	26/147 (17.7%)	5/96 (5.2%)
**Molecular Swabs**				
Total (%)	2859 (18.9%)	2405 (15.9%)	5177 (34.2%)	4714 (31.1%)
Mean for patient (SD)	1.33 (0.59)	1.34 (0.68)	1.50 (0.92)	1.37 (1.21)

SD, standard deviation.

**Table 4 healthcare-10-00205-t004:** Univariate and multivariate analysis in the second pandemic wave for HCWs (Healthcare Workers).

Healthcare Workers	Second Wave October–December 2020 ^1^					
	Univariate				Multivariate	
	Negative	Positive (%)	*p*-Value ^2^	Effect-Size ^4^	OR (95% CI)	*p*-Value ^3^
**Age**			**0.04**	0.06		0.06
≤44	631	51 (7.5%)			1†	
>44	627	76 (10.8%)			1.44 (0.98–2.12)	
**Gender**			0.6	0.01		0.2
Male	616	66 (9.7%)			1†	
Female	642	61 (8.7%)			0.80 (0.55–1.17)	
**Job Title**			**<0.001**	0.07		**<0.001**
Research, administrative staff and other	343	23 (6.3%)			1†	
Ancillary services	139	17 (10.9%)			1.68 (0.55–1.17)	
Non medical-area	180	13 (6.7%)			0.96 (0.47–1.98)	
Nurse	354	62 (14.9%)			**2.24 (1.23–4.08)**	
Physician	239	12 (4.8%)			0.58 (0.26–1.27)	
**Departments**			0.3	0.03		0.6
Research, administrative and operational services	560	47 (7.7%)			1†	
Clinical care	398	46 (10.4%)			1.24 (0.71–2.17)	
Surgery	298	34 (10.2%)			1.02 (0.54–1.91)	

^1^ Two missing values for departments and three for job title. ^2^
*p*-value referred to a Pearson’s Chi-square test, significance level evaluated at 5%. ^3^ Logistic regression model adjusted for age, gender, job title, departments and molecular swabs. ^4^ Cramer’s V statistic. Bold numbers indicate statistically significant results. † indicate reference category.

**Table 5 healthcare-10-00205-t005:** Univariate and Multivariate analysis in the Second pandemic wave for CPs.

Cancer Patients (CPs)	Second Wave October–December 2020					
	Univariate				Multivariate	
	Negative	Positive (%)	*p*-Value ^1^	Effect-Size ^3^	OR (95% CI)	*p*-Value ^2^
**Age**			0.7	0.06		0.39
≤44	502	28 (5.3%)			1 †	
>44	2792	140 (4.8%)			0.83 (0.54–1.27)	
**Gender**			0.9	0.01		0.57
Male	1600	83 (4.9%)			1 †	
Female	1694	85 (4.8%)			1.10 (0.80–1.51)	
**Number of accesses**			**<0.001**	0.1		**<0.001**
One	2299	50 (2.1%)			1 †	
Two	700	57 (7.5%)			**3.76 (2.55–5.55)**	
Three	174	35 (16.7%)			**9.38 (5.92–14.86)**	
Four or more	121	26 (17.7%)			**10.12 (6.07–16.86)**	

^1^*p*-value referred to a Pearson’s Chi-square test, significance level evaluated at 5%. ^2^ Logistic regression model adjusted for age, gender, number of accesses. ^3^ Cramer’s V statistic. Bold numbers indicate statistically significant results. † indicate reference category.

**Table 6 healthcare-10-00205-t006:** Cox-model for CPs in the second pandemic wave.

Cox-Model for Cancer Patients (CPs)	Second Wave October–December 2020	
	HR ^1^ (95% CI)	*p*-Value
**Number of accesses**		
One	1 †	
Two	**4.13 (2.82–6.04)**	**<0.001**
Three	**10.80 (6.99–16.68)**	**<0.001**
Four or more	**21.04 (13.09–33.82)**	**<0.001**

^1^ Adjusted by age and gender. Bold numbers indicate statistically significant results. † indicate reference category.

**Table 7 healthcare-10-00205-t007:** Cox-model for HCWs (Healthcare Workers) in the Second pandemic wave.

Cox-Model for Healthcare Workers (HCWs)	Second Wave October–December 2020	
	HR ^1^ (95% CI)	*p*-Value
**Job Title**		
Research and admin, staff/other	1 †	
Ancillary services	1.70 (0.88–3.29)	0.117
Non medical-area	0.95 (0.48–1.91)	0.895
Nurse	**2.05 (1.16–3.64)**	**0.014**
Physician	0.60 (0.28–1.29)	0.189

^1^ Adjusted by age, gender and Department. Bold numbers indicate statistically significant results. † indicate reference category.

## Data Availability

The data presented in this study are openly available in Zenodo at https://doi.org/10.5281/zenodo.5582606 (accessed on 17 November 2021).

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
