# Peer review of "A SARS-CoV-2 Infection High-Uptake Program on Healthcare Workers and Cancer Patients of the National Cancer Institute of Naples, Italy"

_healthcare, 2022, doi:10.3390/healthcare10020205_

Round 1

Reviewer 1 Report

The article is a bit confusing.
the pathways of the patient who enters the hospital are not well understood, whether for emergency room or scheduled hospitalization, for procedures or visits. for positive health workers, after the swab waiting for the answer went to work? did he have adequate protections? was it an external infection at the hospital?
The small number of infected in an out-of-hospital setting went into bournout without being a hospital dedicated to the covid (since the positive patient was transferred to another facility). What were the internal or regional or state protocols for managing the surveillance of health workers?

Author Response

Reviewer 1

Comments and Suggestions for Authors

  1. The article is a bit confusing.
    the pathways of the patient who enters the hospital are not well understood, whether for emergency room or scheduled hospitalization, for procedures or visits. for positive health workers, after the swab waiting for the answer went to work? did he have adequate protections? was it an external infection at the hospital?

Thank you for your comment. We modified the Materials and Methods, Sample section in line 112: “According to our SARS-CoV-2 infection prevention policy, the pathways of CPs, who had to be admitted to hospital for scheduled hospitalization, procedures or visits, had to carry out the molecular diagnostic test at least 48 hours before hospitalization. In the event of SARS-CoV-2 positive outcome, the local health authorities were alerted and managed the positive CPs and their contacts.”

Moreover, we modified the Materials and Methods, Sample section in line 100: “HCWs of the National Cancer Institute of Naples who agreed to participate in the surveillance program (signed informed consent obtained) were included. This surveillance program included a periodic sampling through nasopharyngeal molecular swab for SARS-CoV-2.”

  1. The small number of infected in an out-of-hospital setting went into bournout without being a hospital dedicated to the covid (since the positive patient was transferred to another facility). What were the internal or regional or state protocols for managing the surveillance of health workers?

Thank you for your comment. We clarified what the regional protocol for managing the surveillance of health workers provides at line 107: “SARS-CoV-2 positive HCWs did not have access to the hospital and were referred to the territorial healthcare system for testing and contact tracing. HCWs who were in contact with COVID-19 patients or SARS-CoV-2 infected individuals were included in a quarantine program by the territorial healthcare system. They were excluded from working activities until the favorable opinion of the health authorities.”

Reviewer 2 Report

Dear Authors, I have read your manuscript with interest.

The current manuscript titled: "A SARS-CoV-2 infection high-uptake surveillance program on healthcare workers and cancer patients in an Italian Cancer Institute" represents an important analysis of evolving field of Infectious Diseases. There is currently a special attention on the COVID-19. The abstract contains all the necessary information in a concise form. The introduction section is clear and easy to read. It provides the basic overview of the current problem.

In my opinion, these are the adjustments which should be made to increase the value of your manuscript:

  1. In title, change please “In-Stitute” to “Institute”.
  2. Line 32: add please abbreviation for ”SARS-CoV-2”.
  3. Line 36: add please abbreviation for ”COVID-19”.
  4. Line 43: add please abbreviation for ”PCR”.
  5. Line 47: add please comma after ”CI 1.23-4.08”.
  6. Line 86: remove please unnecessary spaces.
  7. Line 92: add please abbreviation for ”IRCCS”.
  8. 2. Sample: Please indicate other important and pathognomonic symptoms of COVID-19, for example, chills, anosmia, ageusia, etc.
  9. Line 98: after „positive” add please „test”.
  10. Line 99: change „SARS-COV-2” to „SARS-CoV-2”.
  11. Line 102: delete „(CPs)”.
  12. Line 107: change „COVID” to „COVID-19”.
  13. In figure 1, change „SARS-CoV-2 PCR Testing” to „SARS-CoV-2 RT-PCR Testing”.
  14. In figure 1, change the font to match the main text of the manuscript.
  15. Line 110: change „SARS-CoV2” to „SARS-CoV-2”.
  16. Line 113: change „SARSCoV-2” to „SARS-CoV-2”.
  17. Line 115: change „Real-time PCR detection” to „Real-Time PCR (RT-PCR) detection”.
  18. Line 117: change „Real-time PCR detection” to „RT-PCR SARS-CoV-2 detection”.
  19. Line 129: between 4-5, and 6-11 put a space.
  20. Line 132: change „Confidence intervals” to „CIs”, you have already abbreviated these words above in the text.
  21. Line 143: delete point after (p=0.5).
  22. Table 1: add abbreviation for TOT.
  23. Line 149: change „Sars-Cov-2” to „SARS-CoV-2”.
  24. Table 2 and 3: add abbreviation for SD.
  25. Why Authors compared Nurses which have permanent contact with COVID-19 patients and Administrative staff which haven’t this contact. Obviously there are differences between groups. Which was the aim?
  26. According to what classification did you divide patients into 2 groups: <44 and >44?
  27. Discussion: change „SARS-CoV2, SARS-Cov2, Sars-CoV-2” to „SARS-CoV-2”, „COVID” to „COVID-19”.
  28. After the Discussions, please describe in detail the conclusions based on your results and highlight the practical relevance of this study.
  29. The manuscript contains punctuation errors, please revise the text.

Good luck!

Author Response

Reviewer 2

Comments and Suggestions for Authors

Dear Authors, I have read your manuscript with interest.

The current manuscript titled: "A SARS-CoV-2 infection high-uptake surveillance program on healthcare workers and cancer patients in an Italian Cancer Institute" represents an important analysis of evolving field of Infectious Diseases. There is currently a special attention on the COVID-19. The abstract contains all the necessary information in a concise form. The introduction section is clear and easy to read. It provides the basic overview of the current problem.

In my opinion, these are the adjustments which should be made to increase the value of your manuscript:

  • In title, change please “In-Stitute” to “Institute”.

Thank you for your notice. We modified the Title in line 4: “Institute”

  • Line 32: add please abbreviation for ”SARS-CoV-2”.

We added abbreviation Severe Acute Respiratory Syndrome Coronavirus 2 (SARS‑CoV‑2) both in Abstract section and Introduction section

  • Line 36: add please abbreviation for ”COVID-19”.

We added abbreviation Coronavirus Disease 19 (COVID-19) both in Abstract section and Introduction section.

Line 43: add please abbreviation for ”PCR”.

We modified the Abstract section in line 42: “(Real-Time Polymerase Chain Reaction, RT-PCR).” and in line 45 “were tested with RT-PCR”.

  • Line 47: add please comma after ”CI 1.23-4.08”.

We added comma after ”CI 1.23-4.08” in the Abstract section in line 50.

  • Line 86: remove please unnecessary spaces.

We removed unnecessary spaces.

  • Line 92: add please abbreviation for ”IRCCS”.

IRCCS states for: Istituto di Ricovero e Cura a Carattere Scientifico. We modified the Materials and Methods section in line 81: “National Cancer Institute of Naples (Istituto di Ricovero e Cura a Carattere Scientifico - IRCCS, Fondazione Pascale)”

  • Sample: Please indicate other important and pathognomonic symptoms of COVID-19, for example, chills, anosmia, ageusia, etc.

We added more symptoms of COVID-19 in the Materials and Methods – Sample section in line 104: “A subject was defined as symptomatic if any of the following symptoms/signs were manifested: fever, cough, dyspnea, chills, anosmia, ageusia, etc.”

  • Line 98: after „positive” add please „test”.

We modified the Materials and Methods section in line 102: “SARS-CoV-2 positive HCWs did not have access to the hospital and were referred to the territorial healthcare system for testing and contact tracing.”

  • Line 99: change „SARS-COV-2” to „SARS-CoV-2”.
    We changed „SARS-COV-2” to „SARS-CoV-2” in line 109.
  • Line 102: delete „(CPs)”.

We deleted “(CPs)”.

  • Line 107: change „COVID” to „COVID-19”.

We changed „COVID” to „COVID-19” in line 117.

  • In figure 1, change „SARS-CoV-2 PCR Testing” to „SARS-CoV-2 RT-PCR Testing”.

We changed „SARS-CoV-2 PCR Testing” to „SARS-CoV-2 RT-PCR Testing” in Figure 1.

  • In figure 1, change the font to match the main text of the manuscript.

We changed the font in Figure 1.

  • Line 110: change „SARS-CoV2” to „SARS-CoV-2”.

We changed „SARS-CoV2” to „SARS-CoV-2” in line 125.

  • Line 113: change „SARSCoV-2” to „SARS-CoV-2”.

We changed „SARS-CoV2” to „SARS-CoV-2” in line 127.

  • Line 115: change „Real-time PCR detection” to „Real-Time PCR (RT-PCR) detection”.

We changed „Real-time PCR detection” to „Real-Time PCR (RT-PCR) detection” in line 128.

  • Line 117: change „Real-time PCR detection” to „RT-PCR SARS-CoV-2 detection”.

We changed „Real-time PCR detection” to „ RT-PCR SARS-CoV-2 detection” in line 130.

  • Line 129: between 4-5, and 6-11 put a space.

We modified the text as you suggested.

  • Line 132: change „Confidence intervals” to „CIs”, you have already abbreviated these words above in the text.

We changed „Confidence intervals” to „CIs” in line 147.

  • Line 143: delete point after (p=0.5).

We modified as follow: “(p=0.80)” in line 157.

  • Table 1: add abbreviation for TOT.

We changed “TOT” to “TOTAL” in Table 1.

  • Line 149: change „Sars-Cov-2” to „SARS-CoV-2”.

We changed „Sars-Cov-2” to „SARS-CoV-2” in line 167.

  • Table 2 and 3: add abbreviation for SD.

We added abbreviation for SD in Tables 2 and 3.

  • Why Authors compared Nurses which have permanent contact with COVID-19 patients and Administrative staff which haven’t this contact. Obviously there are differences between groups. Which was the aim?

In our Institute Nurses have contact with cancer patients and not with COVID-19 patients, for this reason, Administrative staff and Nurses groups had similar exposure to COVID-19.

  • According to what classification did you divide patients into 2 groups: <44 and >44?

We divided patients into 2 age groups (≤44ys and >44ys) because the mean age of the cancer patients was 44ys.

  • Discussion: change „SARS-CoV2, SARS-Cov2, Sars-CoV-2” to „SARS-CoV-2”, „COVID” to „COVID-19”.

We changed „SARS-CoV2, SARS-Cov2, Sars-CoV-2” to „SARS-CoV-2”, „COVID” to „COVID-19” in Discussion section.

  • After the Discussions, please describe in detail the conclusions based on your results and highlight the practical relevance of this study.

We added a Conclusion section.

  • The manuscript contains punctuation errors, please revise the text.

We revised all the manuscript for punctuation errors.

Reviewer 3 Report

It is an interesting research that has been undertaken to examine monitoring activities for the transmission of the SARS-CoV-2 virus among healthcare staff and cancer patients. There are few suggestions for improvement.

  1. The authors should provide a more thorough explanation of why cancer patients were chosen for the research in the introduction. What about other chronic diseases or outpatient groups?
  2. Literature reviews, particularly on the Surveillance program, should be included to provide a more comprehensive understanding.
  3. Because nurses constituted the majority of those infected among health-care workers, the research should also provide sound recommendations.

Author Response

Reviewer 3

It is an interesting research that has been undertaken to examine monitoring activities for the transmission of the SARS-CoV-2 virus among healthcare staff and cancer patients. There are few suggestions for improvement.

  1. The authors should provide a more thorough explanation of why cancer patients were chosen for the research in the introduction. What about other chronic diseases or outpatient groups?

Thank you for your comment, the National Cancer Institute of Naples is a cancer centre where only cancer patients are treated. We modified the Introduction section in line 70: “and patients who needed hospital treatment for diseases other than COVID-19 such as suspected cancer disease.” To better clarify we added the reference 9 (Crispo, A.; Montagnese, C.; Perri, F.; Grimaldi, M.; Bimonte, S.; Augustin, L.S.; Amore, A.; Celentano, E.; Di Napoli, M.; Cascella, M.;, Pignata, S. COVID-19 Emergency and Post-Emergency in Italian Cancer Patients: How Can Patients Be Assisted? Front Oncol. 2020, 10, 1571. doi: 10.3389/fonc.2020.01571.). We specified this issue also at line 37 of the abstract: “where only cancer patients are treated”.

  1. Literature reviews, particularly on the Surveillance program, should be included to provide a more comprehensive understanding.

We added references on surveillance program (30-31):

Calò F, Russo A, Camaioni C, De Pascalis S, Coppola N. Burden, risk assessment, surveillance and management of SARS-CoV-2 infection in health workers: a scoping review. Infect Dis Poverty. 2020 Oct 7;9(1):139. doi: 10.1186/s40249-020-00756-6. PMID: 33028400; PMCID: PMC7538852.

Lahner E, Dilaghi E, Prestigiacomo C, Alessio G, Marcellini L, Simmaco M, Santino I, Orsi GB, Anibaldi P, Marcolongo A, Annibale B, Napoli C. Prevalence of Sars-Cov-2 Infection in Health Workers (HWs) and Diagnostic Test Performance: The Experience of a Teaching Hospital in Central Italy. Int J Environ Res Public Health. 2020 Jun 19;17(12):4417. doi: 10.3390/ijerph17124417. PMID: 32575505;

  1. Because nurses constituted the majority of those infected among health-care workers, the research should also provide sound recommendations.

We added a Conclusion section in which we provide practical recommendations and we added the following references regarding Telemedicine approach:

Crispo, A.; Bimonte, S.; Porciello, G.; Forte, C.A.; Cuomo, G.; Montagnese, C.; Prete, M.; Grimaldi, M.; Celentano, E.; Amore, A.; de Blasio, E.; Pentimalli, F.; Giordano, A.; Botti, G.; Baglio, G.; Sileri, P.; Cascella, M.; Cuomo, A. Strategies to evaluate outcomes in long-COVID-19 and post-COVID survivors. Infect Agent Cancer. 2021, 16, 62. doi: 10.1186/s13027-021-00401-3.

Cascella, M.; Marinangeli, F.; Vittori, A.; Scala, C.; Piccinini, M.; Braga, A.; Miceli, L.; Vellucci, R. Open Issues and Practical Suggestions for Telemedicine in Chronic Pain. Int J Environ Res Public Health. 2021;18(23):12416. doi: 10.3390/ijerph182312416.

Cascella, M.; Miceli, L.; Cutugno, F.; Di Lorenzo, G.; Morabito, A.; Oriente, A.; Massazza, G.; Magni, A.; Marinangeli, F.; Cuomo, A.; on behalf of the DELPHI Panel. A Delphi Consensus Approach for the Management of Chronic Pain during and after the COVID-19 Era. Int. J. Environ. Res. Public Health 2021, 18, 13372. https://doi.org/10.3390/ijerph182413372

Reviewer 4 Report

I thank the editor for giving me the opportunity to review this interesting study by Italian colleagues on the SARS-COV-2 infection rate in Health Care Workers (HCWs) and Cancer Patients (CPs) in a single Italian Cancer Institute. The study is well designed and appropriate, so it deserves to be published.

However, I have two major concerns that need to be addressed in order to move towards publication. The first one is to highlight differences between the positive detection of SARS-COV-2 and clinical manifestation of the disease, both in HCWs and in CPs; according to the reference cited by Buonaguro FM et al (9), among the HCWs only 1 patient developed a picture of respiratory severe distress, while 3 cancer patients became symptomatic to require a transfer to a COVID-19 area of ​​the Institution. The second major concern is the request for a greater precision regarding the percentages of HCWs and CPs that have been tested compared to the totality of total HCWs and CPs. In particular, the rate of CPs seems to be extremely low compared to the totality of cancer patients hospitalized in the same study period, therefore it requires at least a minimum mention in the limits section.

Point by point analysis:

Line 43: Please add the percentages of HCWs and CPs tested to the total HCWs and CPs admitted to hospital.

Lines 44-46: please add "of patients tested" when talking about the positivity rate between HCWs and CPs

Lines 48-51: Authors must remain more factual; in the Conclusions section it can be said that: a) the positivity rate between the two waves in the HCWs increased over time, but not in the CPs. b) nurses are at greater risk than administrative, c) it is possible only to SUGGEST that the adoption of safety measures in the HCWs, even if this is not directly related to the positivity of the CPs. Please revise this concept also in the Discussion section.

Lines 72-73: Patients with cancer are not at increased risk of SARS-COV-2 infection, but are patients at increased risk of becoming ill with more serious symptoms of SARS-COV-2 infection (as also reported in the Reference of Liu C et al).

Title: “In-Stitute” is it intentional or is it a grammatical error?

Lines 104-105: how can Authors explain that the CPs had two different methods of detecting SARS-COV-2 infection and were they statistically compared? Please explain this point in the methods or report as a limit.

Figure 1: How many HCWs resulted symptomatic at screening? What about CPs? In the text Authors explain this point only for symptomatic HCWs, not for CPs. Please add an exclusion box to “surveillance program” and “triage” to better magnify this point.

Figure 1: what about HCWs and CPs who refused to participate? An important exclusion box must be reported; the number and percentage must be entered

Line 113: SARS-COV-2 is mis-spelled, please correct

Line 137: 1510 HCWs and 12401 CPs; the data requires an explanation regarding the percentage compared to total HCWs and CPs

Line 143: it is necessary to add how many HCWs and how many CPs were clinically symptomatic and the degree of symptoms, also adding a simple graph, in line with what is also reported by the Italian authorities (Istituto Superiore di Sanità - ISS) regarding the severity clinical stratification of patients reported as positive for SARS-COV-2

Line 144-145: Please explain that “%” is referred to the percentage of total HCWs and CPs. Please reduce the table Font by 1-2 points, to allow a better and more immediate understanding of the table.

Table 2: enter the data (tot /%) of clinical symptomatic patients

Lines 157-158: Please enter that “%” refers to the CPs rate of total hospital admissions

Table 3: enter the data (tot /%) of clinical symptomatic patients

Table 5: the table is high lacking: Authors should insert some information and stratification concerning prognostic patients severity clinical status (ECOG scale, cachexia, type of cancer, etc ...)?

Author Response

Reviewer 4

Comments and Suggestions for Authors

I thank the editor for giving me the opportunity to review this interesting study by Italian colleagues on the SARS-COV-2 infection rate in Health Care Workers (HCWs) and Cancer Patients (CPs) in a single Italian Cancer Institute. The study is well designed and appropriate, so it deserves to be published.

However, I have two major concerns that need to be addressed in order to move towards publication.

  1. The first one is to highlight differences between the positive detection of SARS-COV-2 and clinical manifestation of the disease,both in HCWs and in CPs; according to the reference cited by Buonaguro FM et al (9), among the HCWs only 1 patient developed a picture of respiratory severe distress, while 3 cancer patients became symptomatic to require a transfer to a COVID-19 area of ​​the Institution.

In our study, we did not collect data regarding clinical manifestation of the disease, both in HCWs and in CPs; we cited Buonaguro FM et al as it was the first study that reported the preliminary surveillance program for HCWs and CPs of the National Cancer Institute of Naples.

  1. The second major concern is the request for a greater precision regarding the percentages of HCWs and CPs that have been tested compared to the totality of total HCWs and CPs. In particular, the rate of CPs seems to be extremely low compared to the totality of cancer patients hospitalized in the same study period, therefore it requires at least a minimum mention in the limits section.

We thank the reviewer comment to give us the possibility to revise the analysis to clarify the totality of HCW and CPs in each study period (line 189) of Methods Section (statistical analysis).

Point by point analysis:

3.Line 43: Please add the percentages of HCWs and CPs tested to the total HCWs and CPs admitted to hospital.

We added the following sentence:” 79.7% and 91.7% were the percentage of HCWs and 24.6% and 39.6% of CPs tested with RT-PCR for the detection of SARS-CoV-2 according to the first and the second wave, respectively.”

Lines 44-46: please add "of patients tested" when talking about the positivity rate between HCWs and CPs

We added “of patients tested” as the reviewer suggested in line 48.

  1. Lines 48-51: Authors must remain more factual; in the Conclusions section it can be said that: a) the positivity rate between the two waves in the HCWs increased over time, but not in the CPs. b) nurses are at greater risk than administrative, c) it is possible only to SUGGEST that the adoption of safety measures in the HCWs, even if this is not directly related to the positivity of the CPs. Please revise this concept also in the Discussion section.

Thank you for your comment. We modified the Conclusion section in the abstract and also the Discussion section.

  1. Lines 72-73: Patients with cancer are not at increased risk of SARS-COV-2 infection, but are patients at increased risk of becoming ill with more serious symptoms of SARS-COV-2 infection (as also reported in the Reference of Liu C et al).

We modified Introduction section in line 78: ”such as cancer patients at increased risk of becoming ill with more serious symptoms of SARS-CoV-2 infection”

  1. Title: “In-Stitute” is it intentional or is it a grammatical error?

Thank you for your notice. We modified the Title in line 4: “Institute”

  1. Lines 104-105: how can Authors explain that the CPs had two different methods of detecting SARS-COV-2 infection and were they statistically compared? Please explain this point in the methods or report as a limit.

We deleted the sentence in which we mention the serological test for CPs because we did not analyse these data and we did not include them in our analysis.

  1. Figure 1: How many HCWs resulted symptomatic at screening? What about CPs? In the text Authors explain this point only for symptomatic HCWs, not for CPs. Please add an exclusion box to “surveillance program” and “triage” to better magnify this point.

We did not collect information on symptoms about HCWs and CPs. However we thank the reviewer for the comment and we added the following sentences:

“SARS-CoV-2 positive HCWs did not access to the hospital and were referred to the territorial healthcare system for testing and contact tracing” in line 107 and “In the event of SARS-CoV-2 positive outcome, the local health authorities were alerted and managed the CPs positive and their contacts” in line 114, for HCWs and CPs respectively.

  1. Figure 1: what about HCWs and CPs who refused to participate? An important exclusion box must be reported; the number and percentage must be entered

We reported the exclusion box only for HCWs in Figure 1. All CPs were included in the SARS-COV-2 Infection Prevention because they needed to be admitted to hospital.

  1. Line 113: SARS-COV-2 is mis-spelled, please correct

We corrected “SARS-COV-2 “ in all the manuscript.

  1. Line 137: 1510 HCWs and 12401 CPs; the data requires an explanation regarding the percentage compared to total HCWs and CPs

We agree with reviewer comment and we modified the results section to better explain the data from line 151.

  1. Line 143: it is necessary to add how many HCWs and how many CPs were clinically symptomatic and the degree of symptoms, also adding a simple graph, in line with what is also reported by the Italian authorities (Istituto Superiore di Sanità - ISS) regarding the severity clinical stratification of patients reported as positive for SARS-COV-2

We did not collect information on symptoms about HCWs and CPs.

  1. Line 144-145: Please explain that “%” is referred to the percentage of total HCWs and CPs. Please reduce the table Font by 1-2 points, to allow a better and more immediate understanding of the table.

We added the percentage of total HCWs and CPs in Table 1 and we modified the table font.

  1. Table 2: enter the data (tot /%) of clinical symptomatic patients

We did not collect information on symptoms about HCWs and CPs.

  1. Lines 157-158: Please enter that “%” refers to the CPs rate of total hospital admissions

All SARS-CoV-2 CPs negative were admitted at the hospital; to better clarify we deleted the word “access” from Table 3.

  1. Table 3: enter the data (tot /%) of clinical symptomatic patients

We did not collect information on symptoms about HCWs and CPs.

  1. Table 5: the table is high lacking: Authors should insert some information and stratification concerning prognostic patients severity clinical status (ECOG scale, cachexia, type of cancer, etc ...)?

Our study aimed to analyse SARS-CoV-2 infection among suspected cancer patients admitted at our hospital. We did not collect information on clinical status on CPs.
